

# The evidence of porcine hemagglutinating encephalomyelitis virus induced nonsuppurative encephalitis as the cause of death in piglets

Zi Li[1,*], Wenqi He[1,*], Yungang Lan[1], Kui Zhao[1], Xiaoling Lv[1], Huijun Lu[2], Ning Ding[1], Jing Zhang[1], Junchao Shi[1], Changjian Shan[1] and Feng Gao[1]

[1] Key Laboratory of Zoonosis, Ministry of Education, College of Veterinary Medicine, Jilin University, Jilin, China
[2] Key Laboratory of Zoonosis, Ministry of Education, Institute of Zoonosis, Jilin University, Jilin, China
[*] These authors contributed equally to this work.

Corresponding author
Feng Gao, gaofeng@jlu.edu.cn

## ABSTRACT

An acute outbreak of porcine hemagglutinating encephalomyelitis virus (PHEV) infection in piglets, characterized with neurological symptoms, vomiting, diarrhea, and wasting, occurred in China. Coronavirus-like particles were observed in the homogenized tissue suspensions of the brain of dead piglets by electron microscopy, and a wild PHEV strain was isolated, characterized, and designated as PHEV-CC14. Histopathologic examinations of the dead piglets showed characteristics of non-suppurative encephalitis, and some neurons in the cerebral cortex were degenerated and necrotic, and neuronophagia. Similarly, mice inoculated with PHEV-CC14 were found to have central nervous system (CNS) dysfunction, with symptoms of depression, arched waists, standing and vellicating front claws. Furthmore, PHEV-positive labeling of neurons in cortices of dead piglets and infected mice supported the viral infections of the nervous system. Then, the major structural genes of PHEV-CC14 were sequenced and phylogenetically analyzed, and the strain shared 95%–99.2% nt identity with the other PHEV strains available in GenBank. Phylogenetic analysis clearly proved that the wild strain clustered into a subclass with a HEV-JT06 strain. These findings suggested that the virus had a strong tropism for CNS, in this way, inducing nonsuppurative encephalitis as the cause of death in piglets. Simultaneously, the predicted risk of widespread transmission showed a certain variation among the PHEV strains currently circulating around the world. Above all, the information presented in this study can not only provide good reference for the experimental diagnosis of PHEV infection for pig breeding, but also promote its new effective vaccine development.

## INTRODUCTION

Porcine hemagglutinating encephalomyelitis virus (PHEV) belongs to the order *Nidovirales*, family *Coronaviridae*, and genus *Coronavirus*, and causes encephalomyelitis or vomiting and wasting disease in suckling piglets (*Andries & Pensaert, 1981*; *Mengeling, Boothe & Ritchie, 1972*). Previous studies have demonstrated that the virus has a strong tropism

for the upper respiratory tract and is propagated through the neural route (*Andries & Pensaert, 1980*). The disease caused by PHEV was first reported in Canada in 1958 (*Roe & Alexander, 1958*), and the pathogen was first isolated from the brains of suckling piglets with encephalomyelitis in 1962 (*Greig et al., 1962*). Since then, the infection has been reported in the United States, Japan, Argentina, Belgium, South Korea, China and other pig-raising countries (*Gao et al., 2011*; *Hirano & Ono, 1998*; *Pensaert & Callebaut, 1974*; *Quiroga et al., 2008*; *Rho et al., 2011*; *Sasseville et al., 2001*). Today, many serological surveys have revealed that PHEV is widespread, and there are frequent subclinical infections (*Li et al., 2013*).

In China, PHEV infection first occurred in Beijing in 1985, and the outbreaks caused enormous economic losses for the pig industry (*Chen et al., 2012*; *Dong et al., 2014*; *Gao et al., 2011*). Here we report that there is a suspected outbreak of PHEV infection on a farm in Changchun of Jilin Province, in 2014, resulting in serious economic losses. Many infected piglets characterized with vomiting and nerve symptoms, and some cases were accompanied by screaming or diarrhea; all of the piglets with clinical symptoms died finally. In this paper, the diagnosis was made on the basis of pathologic features, immunohistochemistry, microbiological detection, and RT-PCR. A PHEV field strain was isolated from the brain tissue of infected piglets, and the major structural proteins of the strain were sequenced to identify genetic relationships with other coronaviruses of the genus *Betacoronavirus*.

## MATERIALS & METHODS

### Sample collection and testing

On March 2014, there was an acute outbreak of suspected porcine hemagglutinating encephalomyelitis in suckling pigs on a farm with a total of 502 sows in Changchun, Jilin province, China. At the time of the outbreak, these pigs had not been immunized with any PHEV vaccines; the total proportion of deaths in piglets that had not been weaned was 46.7% (140 dead piglets). The collected samples were tested for PHEV using real-time reverse transcription-polymerase chain reaction (RT-PCR) targeting the HE gene, as well as for other viruses that cause similar clinical symptoms among swine, including porcine epidemic diarrhea virus (PEDV), porcine transmissible gastroenteritis virus (TGEV), porcine deltacoronavirus (PDCoV), and pseudorabies virus (PRV). All experiments on piglets research were performed in accordance with Animal Welfare Ethical Committee of Jilin University guidelines and regulations (permission number 2012-CVM-12). The involved RT-PCR primers were designed based on the most conserved segment of their genomes (Table 1), and subsequently validated by BLAST (http://www.ncbi.nlm.nih.gov/BLAST) with sequences from GenBank. The original samples were diluted 10-fold with phosphate-buffered saline (PBS) and were centrifuged at $3,000\times$ g at $4\,^{\circ}$C for 10 min. The supernatant was filtered through a $0.22$-$\mu$m syringe filter, and was used as inoculums for BALB/c mice or for virus isolation in Neuro-2a cell culture.

### Histopathologic examination

Postmortem examinations were performed and samples submitted for histopathologic examination, including tissues from the brain, heart, spleen, liver, kidneys, and lungs. Paraffin-embedded sections of brain that had characteristic microscopic lesions were

**Table 1** Primer sets used for RT- PCR to differential diagnosis.

| Virus | Primers sequence (5′–3′) | GenBank No | Tm | Gene | Fragment (bp) |
|---|---|---|---|---|---|
| PHEV | TACTGAAACCATTACCACT CTATAACTATGACCGCGAC | AY078417.1 | 56 | HE | 509 |
| PEDV | GAAATAACCAGGGTCGTGGA GCTCACGAACAGCCACA | DQ355221.1 | 55.3 | N | 492 |
| TEGV | GATGGCGACCAGATAGAAGT GCAATAGGGTTGCTTGTACC | AF302264.1 | 58 | N | 612 |
| DPCoV | CGCGTAATCGTGTGATCTATGT CCGGCCTTTGAAGTGGTTAT | KJ569769 | 57.4 | M | 541 |
| PRV | CCGGCCTTTGAAGTGGTTAT CGACCTGGCGTTTATTAACCGAGA | M61196.1 | 56 | gH | 355 |

examined by hematoxylin-eosin staining. Selected paraffin sections for PHEV antigen detection by immunohistochemistry (IHC) tests were treated with normal goat serum for 1 h and anti-HEV 67N monoclonal antibody (*Chen et al., 2012*) (diluted 1:100) overnight; then, the staining procedure was performed according to the kit instructions.

## Inoculation of BALB/c mice with PHEV strains

Thirty 3-week-old male BALB/c mice were randomly and equally divided into three groups. The mice in group 1 were inoculated with the original filtered brain tissue by the intranasal route, the mice in group 2 were inoculated with HEV 67N (GenBank: AY048917) in the same manner, and the mice in the third group formed a negative control group. The permission to work with laboratory animals was obtained from the Animal Welfare Ethical Committee of the College of Veterinary Medicine, Jilin University, China (permission number 2012-CVM-12). All of the mice experiments were carried out at Bio-Safety Level 2 (BSL-2) facilities at the Key Laboratory of Zoonosis, Ministry of Education, College of Veterinary Medicine, Jilin University. Clinical signs were monitored, and immunofluorescence assay (IFA) was performed. The PHEV monoclonal antibody (diluted 1:500) was used as the primary antibody, and a 1:200 dilution of affinity purified fluorescein-labeled goat anti-mouse IgG was used as the second antibody. Cell staining was examined using a fluorescence microscope.

## Virus isolation and propagation

The Neuro-2a cell line was used to isolate PHEV from the original field and from mouse-passaged PHEV samples. Cultured cells were propagated in Dulbecco's Modified Eagle Medium (DMEM, Gibco, USA) supplemented with 10% heat-inactivated fetal bovine serum (Hyclone, Logan, UT, USA) and 1% antibiotic-antimycotic (Gibco, Grand Island, NY, USA). Briefly, a monolayer of cells was washed twice with 2% DMEM, and then was inoculated with the filtered samples. After adsorption for 1 h at 37 °C in 5% $CO_2$, the cells were washed 3 times, and 2% DMEM was added. The cell cultures were examined daily for cytopathic effect (CPE). When more than 80% CPE was evident in the inoculated cell monolayers, the cells and supernatants were harvested together and used as seed stocks for the next passage. After serial passage, the cell cultures were clarified by centrifugation

at $3,000 \times$ g for 30 min at 4 °C and then were further ultracentrifuged at $20,000 \times$ g for 2 h at 4 °C using an ultracentrifuge. The pellet was resuspended and submitted for virological investigation using electron microscopy (EM). The virus isolate was designated PHEV-CC14.

## Virus titration and purification by plaque assay

The 100% confluent Neuro-2a cells in 6-well plates were used for plaque assays of PHEV-CC14 propagation and purification. Briefly, the wells were inoculated with 10-fold serially diluted virus (0.2 mL/well), followed by adsorption for 1 h at 37 °C in 5% $CO_2$; then, the wells were washed 3 times, and 2 mL of the agarose/MEM mixture (1:1) were added. After the plaques were counted and confirmed, uniform and clear plaques were chosen to inoculate 6-well plates directly. When CPE was observed, the positive clones were harvested, and the viral titers were determined. When the Neuro-2a cells were confluent in 96-well plates, 100 mL of 10-fold dilutions of the purified virus were absorbed for 1 h. Viral CPE was monitored for 5 to 7 days, and virus titers were determined by 50% tissue culture infectious dose ($TCID_{50}$).

## PHEV-CC14 structural gene sequencing and phylogenetic analysis

All five of the main structural protein genes, hemagglutinin-esterase (HE), spike (S), small membrane (E), membrane (M) and nucleocapsid (N), of PHEV in the original specimen, in the BALB/c mice infected with passaged PHEV-CC14, and in cell culture were amplified, cloned and sequenced. All of the primers were designed according the sequence of the HEV 67N genome. Viral RNA was extracted from the brain tissue suspensions of symptomatic piglets using a commercial kit (QIAGEN, Hilden, Germany), and the RNA was quantified using a spectrophotometer (BIO-RAD, USA). The RNA was converted to cDNA by an oligo (dT)-priming strategy, and the genes were amplified using PrimeSTAR MAX DNA Polymerase (TaKaRa, Kyoto, Japan). The purified PCR products were cloned into the pMD18-T vector (TaKaRa, Kyoto, Japan) and were introduced into *E. coli* DH5a by transformation. The recombinant plasmids were extracted and verified by PCR and then were sequenced at Shanghai Sangon Biological Engineering Technology and Services Co., Ltd. (China).

The sequence data were assembled and analyzed using DNASTAR and NCBI BLAST (http://blast.ncbi.nlm.nih.gov/Blast.cgi). The percentage similarities of the nucleotides and amino acids were analyzed using DNAMAN and DNASTAR software. The structural gene sequences and other coronavirus strains sequence were subjected to phylogenetic analysis using the neighbor-joining method in MEGA software, version 6.06.

## Statistical analysis

Statistical analysis was performed with either Student's $t$-test or one-way ANOVA with a Bonferroni post hoc test with software provided by GraphPad Prism version 5. Data were presented as means ± S.E.M. $P$ values of <0.05 were considered statistically significant.
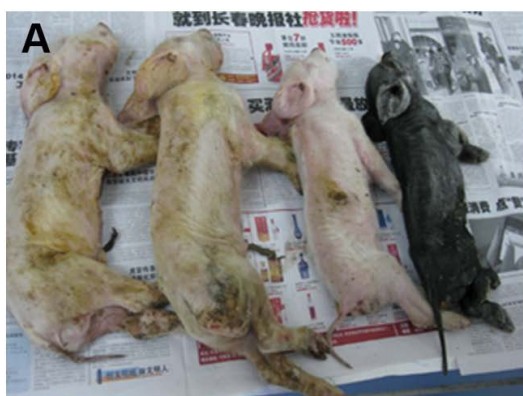
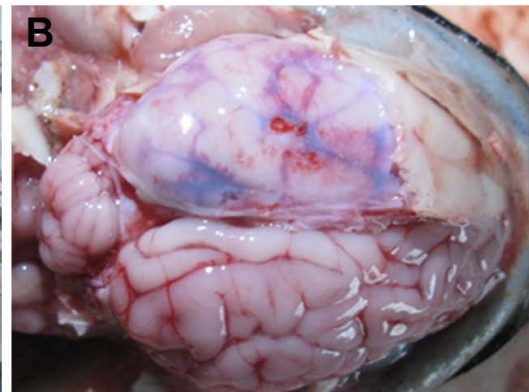

**Figure 1** **Macropathologic images of PHEV infection in piglets from a farm in China.** (A) The dead piglets with vomiting, neurologic symptoms, wasting, and diarrhea. (B) Pathologic autopsy showed congestion or hemorrhage in brain tissue.

# RESULTS

## Pathological examination and pathogen detection

Clinical signs of these suspected infected suckling piglets were consisted of vomiting, diarrhea, wasting, dullness, screaming, anorexia, trembling, and ataxia (Fig. 1A). Pathological examination showed that the main changes in the piglets were congestion, edema and hemorrhage in brain tissue (Fig. 1B). None significant histopathological changes were found in other substantive organs. A total of 54 homogenized tissue suspensions of the brain, spinal cord, lungs, kidneys, spleen and intestinal contents from nine suspected piglets were tested for PHEV by RT-PCR. Of these tested samples, eight of nine brain samples from young nursing pigs on the farms were PHEV positive, as well as eight of nine spinal cord samples and four of nine intestinal content samples (Table 2). Of the 20 PHEV-positive samples, all were negative for PEDV, TGEV, PDCoV, and PRV.

## Histopathologic examination of the PHEV-infected piglets

Postmortem examinations were performed on seven infected piglets for pathologic evaluation. Samples submitted for histopathologic examination included brains from PHEV- infected piglets and antigen-negative piglets. Microscopic examination of brain samples showed characteristics of non-suppurative encephalitis. A large number of glial cells were aggregated to glial nodules in the infected brains (Figs. 2A and 2B). Neurons in the cerebral cortex were degenerated and necrotic, and neuronophagia was widespread (Figs. 2C and 2D). Selected paraffin sections of brain samples that had characteristic microscopic lesions were examined for PHEV antigen by IHC tests with an anti-PHEV monoclonal antibody. In the brains, antigen-positivity in the cytoplasm of nerve cells was distributed widely in the cortical neurons (Fig. 2E). Brain samples from the healthy pig were normal (Fig. 2F).

## Pathogenicity of HEV 67N and PHEV-CC14 in BALB/c mice

Mice in two infected groups were inoculated with HEV 67N and PHEV-CC14, respectively, and were monitored daily for clinical signs of disease. Mice in the HEV 67N-infected group

**Table 2** RT-PCR detection of PHEV and other relevant porcine viruses on tissue samples from nine pigs in Jilin province, China.

| Pig age | Original samples | PHEV and relevant porcine virus detection no. (% positive) | | | | |
|---|---|---|---|---|---|---|
| | | PHEV | PEDV | TEGV | DPCoV | PRV |
| 1-week old | Brain | 6 | – | – | – | – |
| | Spinal cord | 6 | – | – | – | – |
| | IC[a] | 4 | – | – | – | – |
| | Spleen | – | – | – | – | – |
| | Kidneys | – | – | – | – | – |
| | Lungs | – | – | – | – | – |
| < 3-week old | Brain | 2 | – | – | – | – |
| | Spinal cord | 2 | – | – | – | – |
| | IC[a] | – | – | – | – | – |
| | Spleen | – | – | – | – | – |
| | Kidneys | – | – | – | – | – |
| | Lungs | – | – | – | – | – |

**Notes.**
[a] IC, intestinal contents.

showed typical neurological damage, with symptoms of depression, arched waists, standing and vellicating front claws at three days post-inoculation (dpi). The same symptoms occurred in the PHEV-CC14-infected group (Figs. 3A and 3B), but the emergence time was slightly delayed (Fig. 3C, $P < 0.05$). All of the infected mice died within a week, and the mice in the control group survived normally. Paraffin-embedded sections of the infected mouse brain samples were positive for PHEV in the cytoplasm of nerve cells by IFA using a mouse anti-PHEV monoclonal antibody. In the brain, antigen-positive neurons were distributed widely in the cerebral cortex and hippocampus (Fig. 4). In the cerebellum, viral-specific antigen was detected in the Purkinje cells (Fig. 4) but in only a few granular cells.

## Isolation and purification of PHEV-CC14 strain

The Neuro-2a cell monolayer was inoculated with original field and mouse-passaged PHEV-positive samples. At 3 dpi, the inoculated cell monolayer showed visible CPE, in the form of gathering pyknosis and rounded cells that rapidly detached from the monolayer on 4 dpi (Fig. 5A), the mock-inoculated Neuro-2a cells showing normal cells (Fig. 5B). The virus was further serially passed in Neuro-2a cells for a total of 18 passages. Virus growth was confirmed by IFA using the antiserum PHEV, and the antigens were mostly located in the cytoplasm (Fig. 5C). To confirm PHEV replication, viral RNA was extracted from the culture supernatants and was tested by RT-PCR. The presence of PHEV particles in the infected cells was also examined by EM. The EM results showed multiple virus particles approximately 110 to 130 nm in diameter with typical coronavirus morphology (Fig. 5D). Thus, the PHEV strain was successfully isolated and was designated as PHEV-CC14.

Plaque assay was used to plaque isolates and to purify PHEV on Neuro-2a cells, and large clear plaques were evident under an agar overlay medium on the cells. The cloned virus PHEV-CC14 was tested by RT-PCR and was further serially passaged to 20 passages on

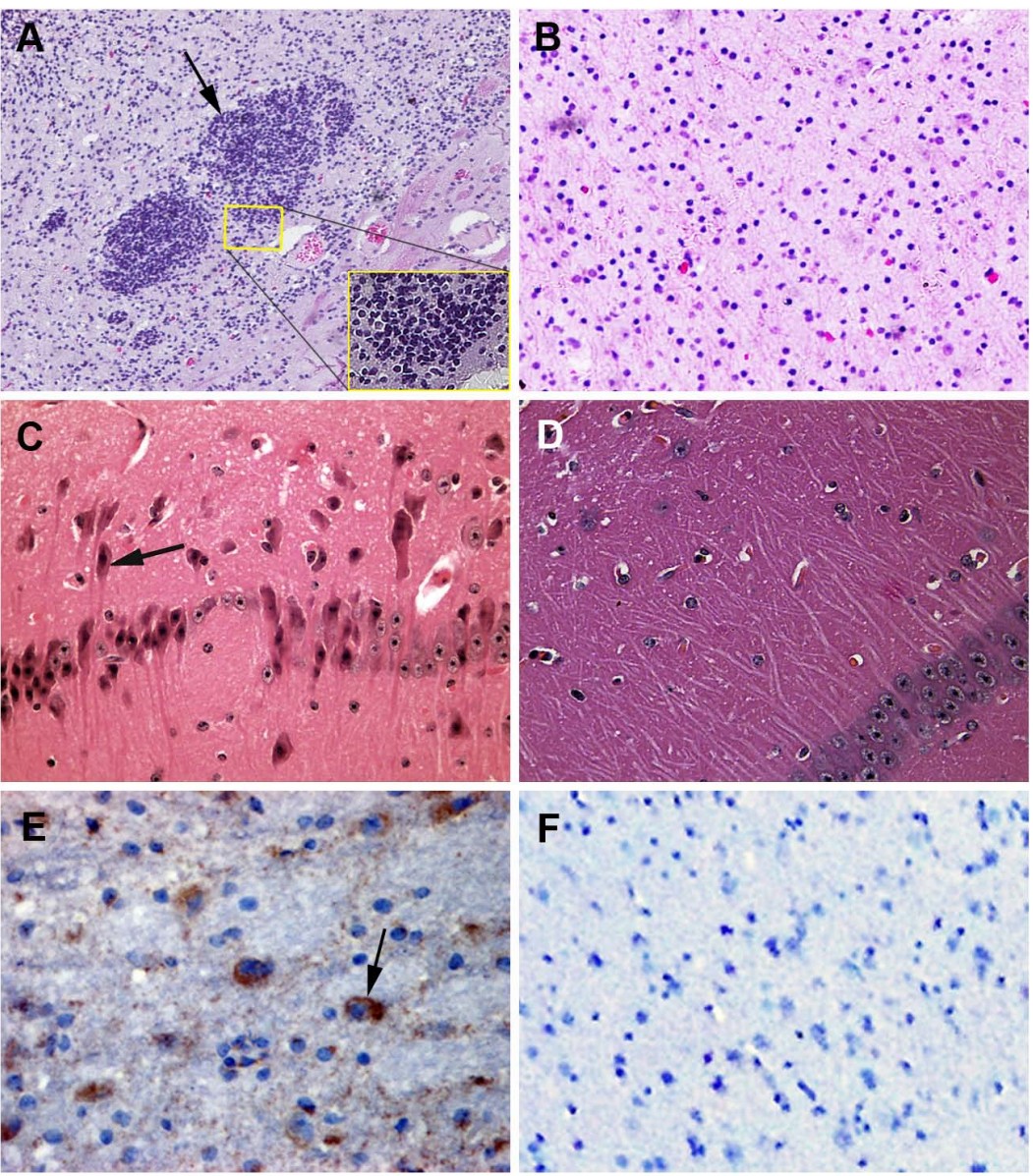

**Figure 2 Samples submitted for histopathologic examination by hematoxylin-eosin staining and IHC assay.** (A) A large number of glial cells aggregating in the affected brains formed glial nodules (arrow); hematoxylin-eosin stain, ×100. (B) Brain samples in the control group were normal; hematoxylin-eosin stain, ×400. (C) Brains from an affected piglet showing PHEV-positive labeling in the cytoplasm of nerve cells (arrows); immunohistochemical staining, ×400. (D) No PHEV-positive labeling of neurons in the negative control group; immunohistochemical staining, ×400. (E) Brains from an infected piglet showing PHEV-positive labeling in the cytoplasm of nerve cells (arrows); immunohistochemical staining, ×400. (F) No PHEV-positive labeling of neurons in the negative control group; immunohistochemical staining, ×400.

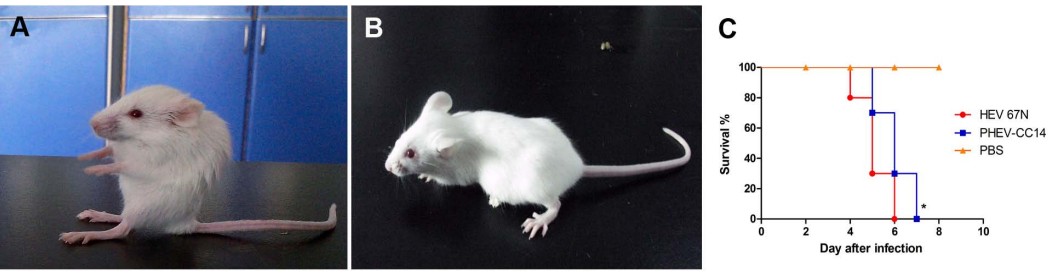

**Figure 3** **Mice experimentally infected with PHEV-CC14.** (A) PHEV-CC14-infected mice showed arched waists, standing and vellicating front claws after at 3 dpi. (B) Mice in the control group survived normally. (C) Survival curves of BALB/c mice. $n = 10$ mice per group; three independent experiments.

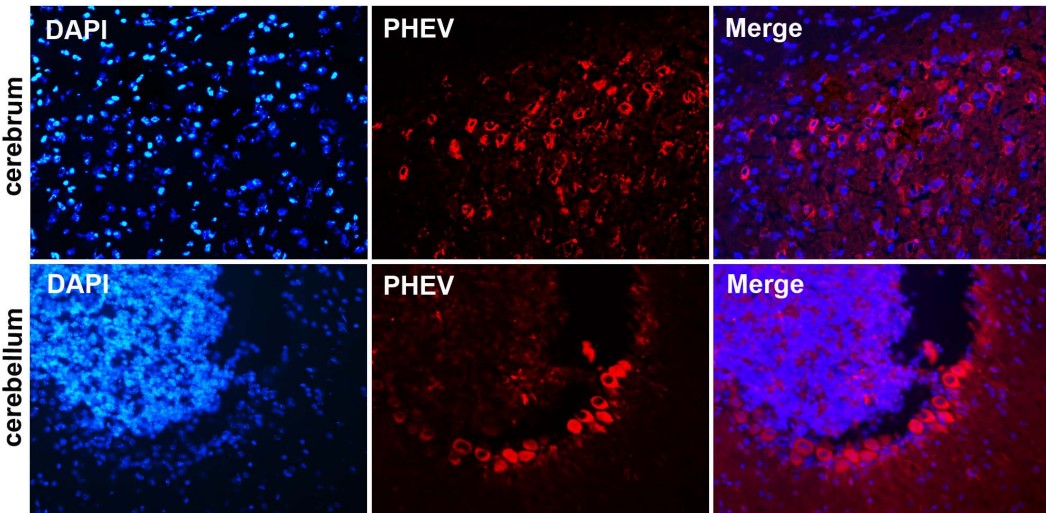

**Figure 4** **Visualization of PHEV-CC14-infected brains from BALB/c mice by immunofluorescent assay using PHEV monoclonal antibody (diluted 1:500).** Immunofluorescent assay in the cerebral cortex, showing large numbers of PHEV-positive neurons (red); original magnification, $\times 100$. PHEV-positive Purkinje cells of the cerebellum were distributed widely (red); original magnification, $\times 400$.

Neuro-2a cells (5.2 $\log_{10}$ PFU/mL). During the serial passages, significant increases in viral RNA titers were observed following each cell passage. The infectious titers of PHEV-CC14 were determined by $TCID_{50}$ and were calculated according to the Reed-Muench method. As shown in Fig. 6, there were no significant differences in replication or proliferation between the PHEV-CC14 and HEV-67N strains in the Neuro-2a cells, but the RNA virus titers in the PHEV-CC14-infected cells ($10^{6.03} TCID_{50}$/mL) were slightly higher than those in the HEV 67N-infected cells ($10^{5.43} TCID_{50}$/mL) after 72 h post-inoculation.

## Sequence and phylogenetic analysis

To examine whether genetic changes occurred in the PHEV-CC14 strain (GenBank: KU127229) compared with other PHEVs available in GenBank, the major structure genes were amplified by specific primers (Table 3) and sequenced. A total of 8,123 nucleotides were determined for strain PHEV-CC14, covering five complete structure genes—HE, S,

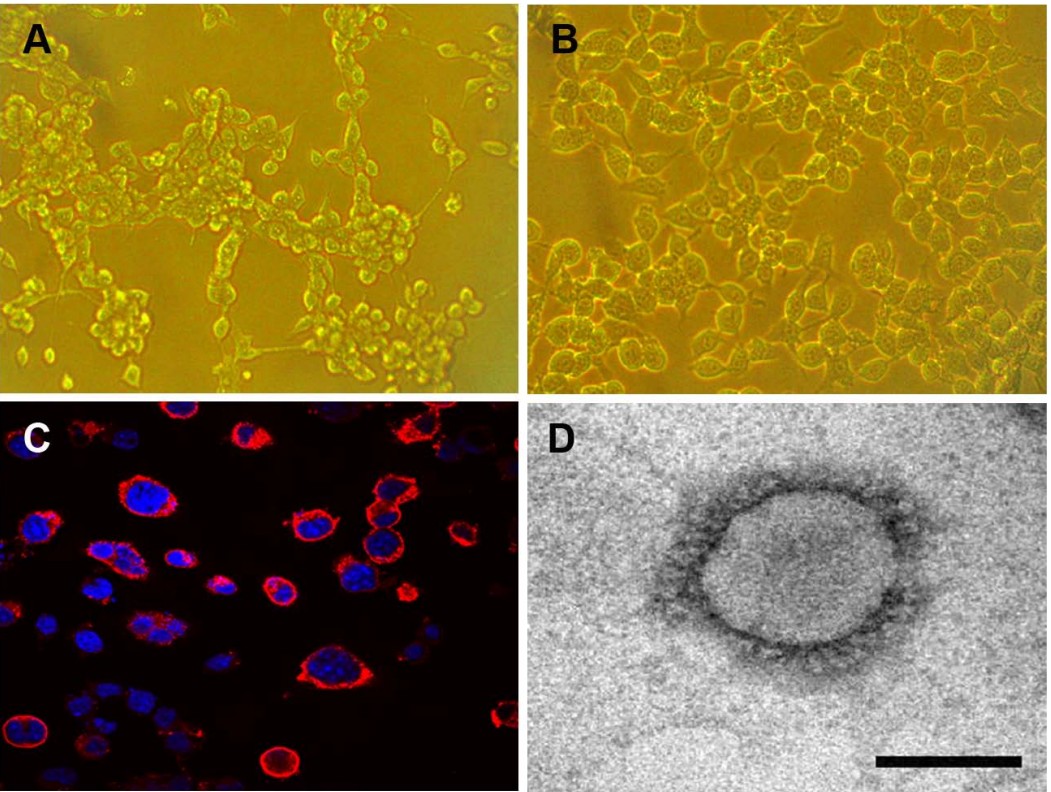

**Figure 5** **Isolation and propagation of PHEV in Neuro-2a cells.** (A) CPE of PHEV-CC14-inoculated Neuro-2a cells at 4 dpi, showing rounded and clustered cells, ×200. (B) Mock-inoculated Neuro-2a cells showing normal cells at 4 dpi, ×200. (C) Detection of PHEV isolate in Neuro-2a cells by immunofluorescent staining using PHEV monoclonal antibody (diluted 1:500), showing immunofluorescent-positive staining mainly evident in the cytoplasm of infected cells (red), ×400. (D) EM of PHEV-CC14-inoculated Neuro-2a cells. Crown-shaped spikes are visible. The samples were negatively stained with 3% phosphotungstic acid . The magnification bar in the picture represents 100 nm in length.

E, M and N—and the locations of the organization of the targeted genes were sketched in a conceptual map (Fig. 7A). Therewith, the corresponding nucleotides and deduced amino acid sequences of the PHEV-CC14 strain were compared with the homologous sequences of PHEVs. The results showed that the PHEV-CC14 strain shared 95%–99.2% nt identities with the other PHEV strains available in GenBank. The structural genes of the PHEV-CC14 strain had the greatest nucleotide sequence similarity (99.2%) to the HEV-JT06 strain (GenBank: ED919227.1), and it shared 99% with HEV 67N (GenBank: AY078417.1). Compared with the HEV 67N strain, there were four nucleotide sense mutations at positions 12 and 114 in the HE gene, 381 in the S gene, 146 in the M gene. These nucleotide changes all induced corresponding amino acid (aa) changes (S12G and T114I in the HE protein; R381H in the S protein; A146T in the M protein). However, residues in the E and N genes of PHEV-CC14 strains were highly conserved in identity with other PHEV reference strains in the GenBank database.

A phylogenetic tree was constructed using the five genes (HE, S, E, M, and N) of PHEV-CC14 with some other PHEV strains obtained from GenBank Database, as well

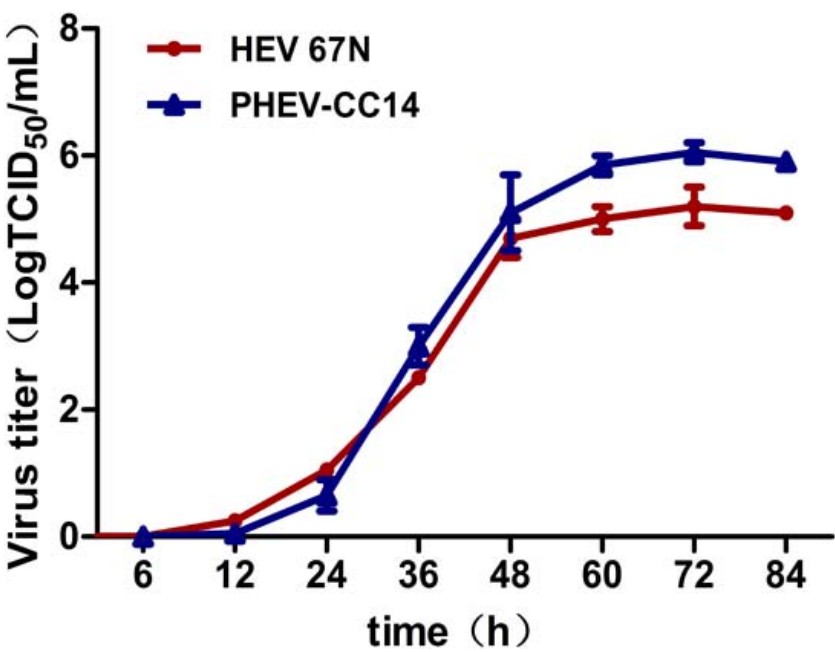

**Figure 6 The growth curves of PHEV strains.** Neuro-2a cells were, respectively, inoculated with PHEV-CC14 and HEV 67N. The $TCID_{50}$ was measured at different time points, and the growth curves were plotted. There was no significant difference in replication or proliferation between the PHEV-CC14 strain and HEV-67N strain ($p > 0.05$).

**Table 3 Sequences of the oligonucleotides for gene-walking RT-PCR.**

| Primers | Primers sequence (5'–3') | Gene | Fragment (bp) |
|---|---|---|---|
| P1/P2 | TACTGAAACCATTACCACT CTATAACTATGACCGCGAC | I | 1,275 |
| P3/P4 | GAAATAACCAGGGTCGTGGA GCTCACGAACAGCCACA | II | 1,614 |
| P5/P6 | GATGGCGACCAGATAGAAGT GCAATAGGGTTGCTTGTACC | III | 1,674 |
| P7/P8 | CGCGTAATCGTGTGATCTATGT CCGGCCTTTGAAGTGGTTAT | IV | 1,390 |
| P9/P10 | CCGGCCTTTGAAGTGGTTAT CGACCTGGCGTTTATTAACCGAGA | V | 256 |
| P11/P12 | ATGAGTAGTCCAACTACAC TATTTCTCAACAATGCGGTGTC | VI | 685 |
| P13/P14 | TCAGGCATGGACACCGCATT AGAGTGCCTTATCCCGACTTT | VII | 1,463 |
| P15/P16 | TTACAGCACTTAGATCACGTAGAT TAAACTCTGGCTTCGCCAGGTAAT | VIII | 2,195 |

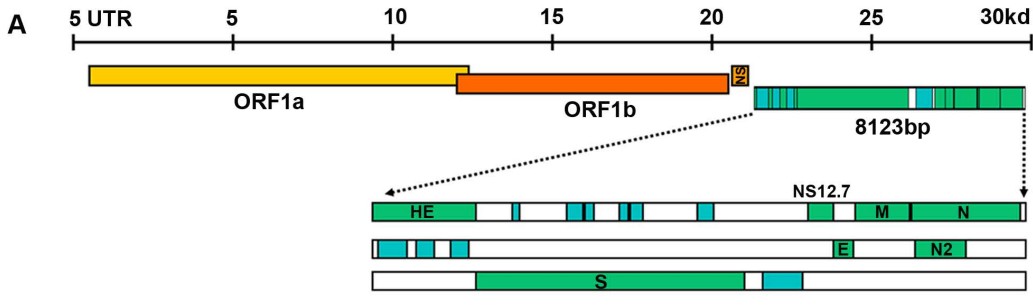

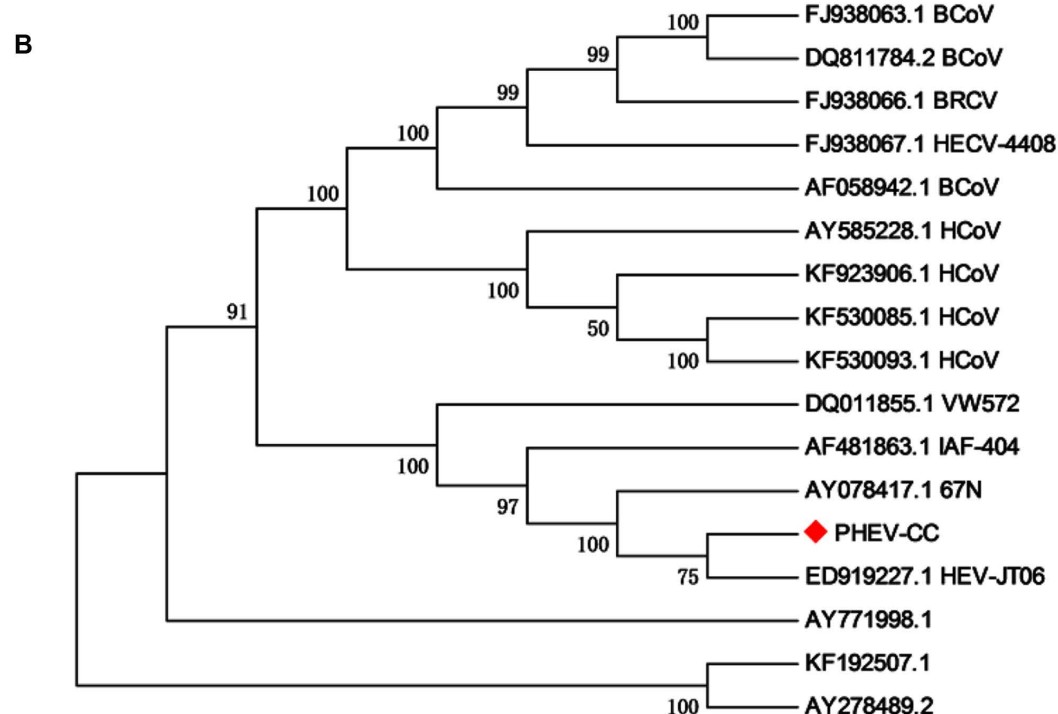

**Figure 7 Protein structure prediction and phylogenetic analysis.** (A) Schematic illustration of the organization of the targeted genes coding for the five structural proteins, consisting of PHEV HE, S, E, M and N genes (reference virus HEV-67N). (B) Phylogenetic analyses based on amino acid sequences of the five major structural proteins from PHEV in this study (indicated with triangle) and other published PHEV sequences, as well as related coronaviruses. Reference sequences obtained from GenBank are indicated by strain names and accession numbers. The trees were constructed using the neighbor-joining method in MEGA software, version 6.06. Bootstrap analysis was performed on 1,000 trials, and values are indicated adjacent to the branching points.

as several members of the coronaviruses (Fig. 7B). Phylogenetic analysis of the five genes clearly showed that the PHEV-CC14 strain clustered into a subclass with a HEV-JT06 strain from China isolated in 2006, and a similar finding showed that the PHEV stains in China were highly homologous with a North American strain (AY078417). Additionally, the homology of the deduced amino acid sequences between the PHEV-CC14 strain and HCoV-OC43 (GenBank: KF530085) was as high as 91%.

## DISCUSSION

In March 2014, there was a suspected outbreak of PHEV infection on a farm in Changchun in the Jilin province of China. The clinical signs consisted of vomiting, twitching, wasting, and diarrhea were observed in suckling piglets. The histologic autopsy showed that there were pinpoint petechiae in the kidneys, thinning of the intestinal wall, and hemorrhage in the brain. Due to the great similarity between PHEV and pseudorabies virus (PRV) infection in piglets, it was difficult to distinguish them only by clinical and autopsy symptoms. Therefore, PRV and PHEV were first detected by PCR and RT-PCR, respectively, and the test results showed that PRV was negative, and PHEV was positive, thus excluding PRV infection. At the same time, no sow abortions or stillbirth phenomena were found in the pigs, also supporting the above results. Immunohistochemical staining results further confirmed PHEV infection in the brains of the dead piglets. Vomiting and neurological symptoms are common in piglets infected with PHEV, but the symptom of diarrhea is relatively rare. Because the outbreak was characterized by vomiting and diarrhea, some other viral infections with similar clinical symptoms have been reported in pigs, including PEDV, TGEV and DPCoV (*Ma et al., 2015*; *Song, Moon & Kang, 2015*; *Tanaka et al., 2015*). In this study, these viruses were further detected to exclude misdiagnosis or mixed infection. Thus, the case was identified as simple PHEV infection, and we successfully isolated a PHEV field strain as PHEV-CC14.

PHEV has a typical neural tropism, and it invades the central nervous system via the peripheral nervous system (*Hirano et al., 2004*; *Lafaille et al., 2015*; *Lee et al., 2011*; *Zhang et al., 2014*). Previous studies have shown that the virus successfully killed 1- to 8-week-old mice readily by different routes, and viral antigen was detected in both peripheral nerves and the CNS (*Hirano et al., 2001*; *Hirano et al., 1995*; *Hirano et al., 2006*). In this paper, three-week-old BALB/c mice were chosen for inoculation with PHEV-CC14 by the intranasal route, and the results of the experiment confirmed that PHEV-CC14 had strong pathogenicity to mice.

Viral titers of PHEV-CC14 or HEV 67N were determined by TCID$_{50}$, and the growth curve showed that there was no significant difference in replication or proliferation between them. To characterize the virus isolates, the complete structural genes were sequenced and analyzed, and the phylogenetic relationships among the coronavirus strains were determined. Phylogenetic trees showed that the wild type and HEV-JT06 shared the highest homology, and that identified with HEV 67N was 99%. These findings suggested that PHEV strains currently circulating in China are closely related. Notably, the PHEV strain JT06 isolated from Jilin province, China, in 2006 was most closely related to the emerging PHEV-CC14 strains, suggesting that they could be derived from a similar ancestral strain. Furthermore, we performed sequence alignment and homology analysis between PHEV-CC14 and other coronaviruses, which including MHV, BCoV, HCoV-OC43 and Bat CoV, and we found it shared up to 91% homology with HCoV-OC43 (GenBank: KF530085) (*Gonzalez et al., 2003*; *Li, 2015*; *Snijder, Horzinek & Spaan, 1993*). This finding suggested that, although PHEV infection in humans has not been reported currently, there is a definite potential threat to human health.

According to the deduced amino acids of the PHEV-CC14 strains, genetic evolution and variation analyses were performed. It was found that the broad variation occurred in the encoded structural proteins and functional region. There were five major structural proteins of PHEV, HE, S, E, M and N proteins, encoded from 5′UTR to 3′UTR (*Vijgen et al., 2006*; *Weiss & Navas-Martin, 2005*). Sequence analyses of the five structural proteins of the PHEV-CC14 field isolate suggested that PHEV has remained more genetically stable in the E, M and N proteins (*Schultze et al., 1990*; *Vieler et al., 1995*). The HE proteins of some coronaviruses are involved in the release of virions from the host cell, and it has been shown to have acetylesterase activity and to function as a receptor-destroying enzyme, which might be related to the early adsorption of coronavirus (*Schultze et al., 1991*). Compared with HEV 67N, there were two amino acids (S12G, T114I) in the HE protein of PHEV-CC14 that were meaningful mutations, while six amino acid variations were observed (S12G, S15G, K49N, T114I, V116A, L161F) when blasted with the IAF-404 strain. We hypothesize that the amino acid variation of the HE protein might have a certain effect on replication and virulence, but it was difficult to explain the differences in virulence among the pigs, based on the amino acid changes. In addition, there were some variations of the amino acids in the S protein, which plays vital roles in viral entry, cell-to-cell spread, and the determination of tissue tropism (*Dong et al., 2015*; *Lu, Wang & Gao, 2015*). Therefore, the differences in virulence of PHEV strains might be caused by multiple factors, and the variation of the whole genome has resulted in changes in their antigenic differences.

## CONCLUSIONS

Briefly, the outbreak on the pig farm in northern China was caused by PHEV, and the virus was isolated, systematically characterized and designated PHEV-CC14. This work will enrich the data on the genome and molecular epidemiology of PHEV and will provide material for further study of the virulence of PHEV, which should have a certain theoretical and practical significance.

### Funding

This study was supported by the National Natural Science Foundation of China (Nos: 31472194, 31272530, 31172291), the National Key R&D Program of China (Nos: 2016YFD0500102, 2016YFD0500707), and the Youth Scientific Research Foundation of Jilin Province (No. 20160520033JH). The funders had no role in study design, data collection and analysis, decision to publish, or preparation of the manuscript.

### Grant Disclosures

The following grant information was disclosed by the authors:
National Natural Science Foundation of China: 31472194, 31272530, 31172291.
National Key R&D Program of China: 2016YFD0500102, 2016YFD0500707.
Youth Scientific Research Foundation of Jilin Province: 20160520033JH.

## Competing Interests

The authors declare there are no competing interests.

## Author Contributions

- Zi Li and Wenqi He conceived and designed the experiments, performed the experiments, analyzed the data, wrote the paper, prepared figures and/or tables.
- Yungang Lan, Kui Zhao and Xiaoling Lv conceived and designed the experiments, performed the experiments, analyzed the data.
- Huijun Lu, Ning Ding, Jing Zhang, Junchao Shi and Changjian Shan performed the experiments, contributed reagents/materials/analysis tools.
- Feng Gao analyzed the data, reviewed drafts of the paper.

## Animal Ethics

The following information was supplied relating to ethical approvals (i.e., approving body and any reference numbers):

The permission to work with laboratory animals was obtained from the Animal Welfare Ethical Committee of the College of Veterinary Medicine, Jilin University, China (permission number 2012-CVM-12).

## DNA Deposition

The following information was supplied regarding the deposition of DNA sequences:

GenBank: KU127229; http://www.ncbi.nlm.nih.gov/nuccore/1021673930.

## Data Availability

GenBank: KU127229.

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
