# Peer review of "The evidence of porcine hemagglutinating encephalomyelitis virus induced nonsuppurative encephalitis as the cause of death in piglets"

_PeerJ, doi:10.7717/peerj.2443_

## Round 0.1 · original submission · Minor Revisions

Zi Li et al (peerj-10903) reported clinical and experimental evidences of porcine hemagglutinating encephalomyelitis virus induced nonsuppurative encephalitis as the cause of death in piglets. They did the comprehensive diagnosis of PHE, isolated and characterized the PHEV-CC14 strain, and performed the phylogenetic analysis of its major structural genes. Their finding is significant.

In addition to the comments of the reviewers, specific comments are:

1.All experiments reporting results on animal research must be performed in accordance with relevant institutional and national guidelines and regulations. In the manuscript, authors must identify the full name of the ethics committee that approved the work.

2.please authors add statistical analysis in Material and Methods.

·

Basic reporting

No Comments

Experimental design

No Comments

Validity of the findings

No Comments

Additional comments

The manuscript of Zi Li et al (peerj-10903) was aimed to prove nonsuppurative encephalitis was the main death cause of porcine hemagglutinating encephalomyelitis virus-infected piglets.The authors found that some characteristics of typical non-suppurative encephalitis and PHEV-positive labeling of neurons in dead piglets, and typical symptoms of non-suppurative encephalitis in the mice infected with the isolated PHEV-CC14. Therefore, the authors concluded that porcine hemagglutinating encephalomyelitis virus induced nonsuppurative encephalitis as the cause of death in piglets.

General comments.

This manuscript brings clinical and experimental evidences for supporting nonsuppurative encephalitis as the main death cause of porcine hemagglutinating encephalomyelitis in piglets. The importance and clinical significance of obtained results should be described more directly and in the Abstract section.

Specific comments
1. In Line 20-23, the sentence should be revised.
2. In line 43, “and it” change into “and”.
3. In line55-60. Please use present tense.
4. In table1, please provide the GenBank.NO. for the viral genes.
5. In line 88, please provide the detail information for monoclonal antibody.
6. In line 158,169 and 182, please revise subtitle according to your contents to make these titles look more like the results.

Reviewer 2 ·

Basic reporting

In this study authors identified hemagglutinating encephalomyelitis virus (PHEV) infection in piglets, which caused high mortality. This is a very important study by isolating PHEV strain from piglets and authors performed careful experiments to characterize PHEV using multiple approaches including cell culture, sequencing, histo and immunohistologies.

Experimental design

This study is original by isolating PHEV virus from infected piglets. Experiments were well designed and performed. In particular, the virus was isolated and identified through serial experiments including cell culture and inoculating in mice.
However, it is not clear whether the experiments on piglets were also approved by institutional animal care committee. This should be addressed.

Validity of the findings

Finding was solid since the virus was isolated from infected piglets and replicated in cells and mice. Moreover, sequencing was performed to validate the virus strain.

Additional comments

Authors should address whether experiments involved in piglets were also approved by institutional animal use and care committee.

Reviewer 3 ·

Basic reporting

The manuscript titled with "The evidence of porcine hemagglutinating encephalomyelitis virus induced nonsuppurative encephalitis as the cause of death in piglets", its structure conforms to PeerJ Standard, its figures are relevant, high quality, and well labelled & described.

Experimental design

No Comments

Validity of the findings

Data in this paper is robust, statistically sound and controlled.

Additional comments

This paper carried out the comprehensive diagnosis of PHE, isolated and characterized the PHEV-CC14 strain, and performed the phylogenetic analysis of its major structural genes (proteins). It provided good reference for the experimental diagnosis of PHEV infection for pig breeding. Furthermore, it showed molecular mutation of certain structural genes of isolated virus and promoted its new effective vaccine development.

I think it should be in the scope of our journal and could accept it after language polishing.

The following is the minor suggestion on the paper.

Introduction
Line 43, Nidoviracles—Nidovirales

Materials & Methods
Line100-101, the second conjugated antibody should be “goat anti mouse IgG” for your first antibody is McAb.

Results &Conclusion
Line 149-150, Pathological anatomy—Pathological examination
Line 172, “HEV 67N-infected group” is better in the paper

Question

Figure 2
The pictures is the histopathological and immunochemical examination of the clinical sample, so in the legend it is better no use of the control group, how do you think the healthy pig or other words. Furthermore, it is better to use more pictures of higher magnitude to demonstrate the damage of neuron and the morphology of glial cells (non-suppurative encephalitis)

---

## Round 0.2 · accepted · Accept

Thank you for your revised manuscript which appropriately addresses the reviewer comments.